# The Manufacturing Process of Kiwifruit Fruit Powder with High Dietary Fiber and Its Laxative Effect

**DOI:** 10.3390/molecules24213813

**Published:** 2019-10-23

**Authors:** Ziqi Zhuang, Min Chen, Jihan Niu, Na Qu, Bing Ji, Xiang Duan, Zhande Liu, Xuebo Liu, Yutang Wang, Beita Zhao

**Affiliations:** 1College of Food Science and Engineering, Northwest A&F University, Yangling, Shaanxi 712100, China; 906881422@nwafu.edu.cn (Z.Z.); 18794419539@163.com (M.C.); NiuJiHan0716@163.com (J.N.); duanxiang402@163.com (X.D.); xueboliu@aliyun.com (X.L.);; 2Patent examination cooperation Jiangsu center of the patent office, Su Zhou, Jiangsu 215000, China; japannlhquna@126.com; 3College of Horticulture, Northwest A&F University, Yangling, Shaanxi 712100, China; dezhanliu@vip.sina.com

**Keywords:** kiwifruit, microencapsulate, dietary fiber, ultra-micro kiwifruit powder, constipation, laxative effect

## Abstract

Kiwifruit is rich in vitamins, minerals, dietary fiber and other functional components, and it has long been used as a functional food to treat intestinal ailments such as constipation. The current research made full use of the kiwifruit, the juice was prepared by microencapsulation, and the dietary fiber in kiwifruit pomace was modified by enzymatic hydrolysis and grinding, then, the two were mixed to obtain an ultra-micro kiwifruit powder (UKP). In addition, the laxative effect of the UKP was verified by a diphenoxylate induced constipated mice model. The results demonstrated that compared with the raw samples, the retention rate of vitamin C, lutein and catechin in UKP were 83.3%, 81.9% and 88.3%, respectively, thus effectively avoiding the loss of functional components during the processing of kiwifruit. Moreover, α-amylase, protease and the ball milling process effectively reduced the size of dietary fiber in kiwifruit pomace, and its water-holding capacity (WHC), oil-holding capacity (OHC) and swelling capacity (SWC) were enhanced by 1.26, 1.65 and 1.10 times, respectively. Furthermore, to analyze the laxative effect of the UKP, a constipation mice model was established by diphenoxylate treatment (5 mg·kg^−1^, i.g.) for the last week, with or without UKP supplementation (2.4 g·kg^−1^ B.W. per day) for 4 weeks. The results demonstrated that UKP significantly increased feces condition (fecal output and dejecta moisture content, gut transit (the intestinal propulsion rates) and substance P (SP) levels in portal vein plasma, and it decreased the whole gut transit time and mucinogen granules secreted by goblet cell in constipated mice.

## 1. Introduction

Constipation is a common problem affecting children and adults, and its symptoms often appear as difficulty, discomfort or infrequency of defecation and a sense of incomplete evacuation. It is characterized by long-term retention of feces in the colon and lack of moisture content within the lumen [1]. According to the American Gastroenterological Association, among healthy people, the frequency of bowel movements may vary from three times per week to three times daily [2]. Contributing factors to constipation include gender, age, diet and lifestyle choices, use of certain medications, and bowel habits [3,4,5]. In addition, disorder of the gastrointestinal function is also a major cause of constipation [6]. Therefore, improving intestinal health through diet is an effective way to alleviate constipation. 

Kiwifruit is widely reported as a functional food and a nutraceutical source. Of all the beneficial effects associated with the consumption of green kiwifruit, laxation activity deserves our attention. Recent research has demonstrated that the consumption of kiwifruit (Hayward variety) relieves the symptoms of gastrointestinal (GI) disorders such as indigestion, bloating and constipation, and the daily inclusion of fresh kiwifruit in the diet of the elderly has also been reported to increase the frequency and ease of laxation [7]. Kiwifruit contains a variety of vitamins such as vitamin C and vitamin E, polyphenols, lutein, and minerals, which could maintain gastrointestinal health [8,9]. However, vitamin and polyphenols are highly susceptible to oxidation and epimerization at higher temperatures, and alkaline pH and oxygen can further reduce their activity [10]. Microencapsulation of the bioactive compound using biopolymer could protect the functional components against physical and chemical stress [11,12]. γ-Cyclodextrin (GCD) offers several advantages such as low bulk density, resistance to caking, blandness, excellent mouth-feel, film formation, effective binding and oxygen barrier [13,14]. The incorporation of microencapsulated bioactive compound into foods may have a minimal impact on the qualitative characteristics of the kiwifruit product. Although kiwifruit is rich in active ingredients, it is relatively lacking in dietary fiber and the synergistic effect of the juice and pomace is better for improving constipation.

Many studies have demonstrated that the peel of kiwifruit contains large amounts of dietary fiber (DF), which benefits the physiological activities of humans by maintaining gastrointestinal health [15]. A fiber-rich diet increases the fecal output of water and dry matter, playing a positive role in constipation [16]. It has been reported that reducing the particle size of insoluble dietary fiber (IDF) to a certain micro-scale might enhance their physicochemical properties [17]. In the current research, ball milling was employed to reduce the particle size of pomace IDF and improve its adsorbing capacity [18], aiming to improve its laxative effects in constipation model mice.

## 2. Results 

### 2.1. The Microstructure of Microencapsulated Kiwifruit Juice and Modified DF Pomace 

The product image and morphology of microencapsulated kiwifruit juice and grinded insoluble dietary fiber (IDF) is depicted in Figure 1. The particle morphology of kiwifruit juice obtained by spray drying is shown in Figure 1B,C. It is clear that the structure of the microencapsulation is spherical and non-agglomerated particles with a uniform appearance. The raw kiwifruit pomace showed complete sheet-like morphologies and flaky structure (Figure 1C,D). After milling treatment, a reduction in particle size with more sharp and prominent edges was observed (Figure 1E,F). Figure 1G–I elucidated the fluorescent properties on the microstructure of ultra-micro kiwifruit IDF with an inverted fluorescence microscope. These results revealed that pulverization by ball milling effectively reduces the size of the IDF particles to a submicron scale; and thus, it is feasible to improve the physicochemical properties. 

### 2.2. The Physicochemical Properties of Modified DF Pomace 

The reduction in particle size has a significant effect on the physical structure of DF, which is related to its hydration properties such as water-holding capacity (WHC), oil-holding capacity (OHC), swelling capacity (SWC) and apparent density (AD). These physicochemical properties were studied before and after ultra-micro grinding. As shown in Table 1, WHC and SWC were increased from 3.62 to 4.56 and 6.98 to 7.69, respectively. And that is, kiwifruit DF showed increased hydration properties after the grinding treatment. Similarly, the raw kiwifruit IDF had an OHC of 0.66 g oil/g dry sample. After the grinding process, the value increased to 1.09 g oil/g dry sample. Meanwhile, AD decreased with the decrease in the DF particle size, and was reduced from 0.41 to 0.23 after ultra-micro grinding. The results indicate that the particle size of DF plays an important role in determining its hydration properties, such as WHC and SWC, and oil retention capacity. 

### 2.3. The Chemical Analysis of UKP

On the basis of water solubility, dietary fiber can be divided into insoluble dietary fiber and soluble dietary fiber. The content of dietary fiber before and after grinding is shown in Table 2. The results showed that TDF was increased from 81.27% to 84.48% and IDF was increased from 75.96% to 72.61%, while SDF was increased from 5.31% to 11.87% after grinding, suggesting that ultra-micro grinding causes a restructuring of kiwifruit fiber components from IDF to SDF. The phenolic and vitamin C content in raw kiwifruit and UKP determined by HPLC and 2,6-dichloroindophenol titrimetric method are presented in Table 3. The retention rates of the phenolic and vitamin C were all above 87%. Thus, the active ingredients were basically retained.

### 2.4. UKP Regulates GI Motility and Fecal Output in Mice 

Table 4 shows that no significant differences in body weight and food intake among the four groups of mice were found during the test period, suggesting that the diphenoxylate or UKP did not affect mice motor activities. The model group mice (diphenoxylate, 5 mg·kg^−1^, i.g.) had significantly lower fecal output (FO) and dejecta moisture content (DMC), which shortened the intestinal propulsion rates (IPR) and increased the whole gut transit time (WGT) compared with the control mice (Figure 2). Feeding UKP and freeze-dried kiwifruit (FKD) substantially improved the feces condition (Figure 2A,B) and gut transit (Figure 2C,D) in constipated mice. However, in comparison with freeze-dried kiwifruit, UKP had no significant difference in these four testing indicators. These results further revealed that the kiwifruit, after the encapsulation and modification process, has the potential to promote intestinal peristalsis and alleviate constipation.

### 2.5. Effects of UKP on Preventing Intestinal Damage 

Hematoxylin and eosin (H&E) staining was used to examine histopathological changes in the ileum and colon. As shown in Figure 3, there were no alterations in villus height and crypt depth in the jejunum in the four groups. However, the mucinogen granule secreted by goblet cells of intestinal mucosa was significantly increased after diphenoxylate treatment in the model group compared with the control group. Also, the goblet cell secretion was significantly decreased in the UKP treated group.

### 2.6. Effects of UKP on Expressions of Portal Vein Serum Neurotransmitter

As shown in Figure 4A, the lowest level of substance P (SP) was detected in portal vein plasma of model group mice, while higher levels of SP were observed in the UKP and FKD treatment groups. However, the levels of somatostatin (SS) and vasoactive intestinal peptide (VIP) showed barely any difference after diphenoxylate treatment compared with the control group (Figure 4B,C). It is possible that these changes are caused by the increase in dietary fiber.

## 3. Discussion

The structure of the microencapsulation was clearly spherical with non-agglomerated particles of uniform appearance. Generally, spherical type microcapsules allow powder to flow better because they lack surface roughness and do not form agglomerates [19]. Furthermore, the spheres contained some dents on their surface. This could be due to the instant removal of water during spray drying, which might have caused the collapse of the walls of the spheres [20]. The raw kiwifruit pomace showed complete sheet-like morphologies and was flaky in structure. The reduction in the IDF particle size to micro scale leads to some changes in the structure and surface area and brings some new, outstanding characteristics that bulk fiber does not possess, such as the fluorescent properties. 

Generally, the dramatic decrease in the particle size, increase in the surface area and the alteration of the fiber matrix structure after the grinding process resulted in a significant increase in the porosity and adsorption sites, which enhances its physical entrapment of water and oil. In the absence of a matrix structure, the relative surface area and the total amount of water held by fiber varies inversely with the particle size [21,22,23]. Insoluble fiber with high water holding capacity would increase the stool weight and stimulate bowel path wriggles [24]. This supports a previous viewpoint that UKF could be helpful to relieve constipation.

A SDF content higher than 10% is deemed to be a balanced DF composition [25]. It is worth noting that the SDF content of ultra-micro DF powder in this work was higher than 10%. As expected, microencapsulation of the bioactive compound using γ-cyclodextrin (GCD) effectively protects kiwifruit against physical and chemical stress. In the current research, ball milling was employed to reduce the particle size of pomace IDF and improve its adsorbing capacity, and the results demonstrated that it improved the laxative effects in constipation model mice. Therefore, UKP with high functional DF, phenolic and vitamin C content is helpful to relieve constipation and to lower the incidence of certain types of intestinal diseases. 

Goblet cells (GC), which are specialized intestinal epithelial cells secrete mucinogen granules, predominantly Muc2 and Muc5AC into the intestinal lumen [26]. Mucinogen granules protect the epithelium from dehydration, physical abrasion, and microbial translocation [27]. Therefore, mucin granules can accumulate within the goblet cell cytoplasm because of constipation. It does this by promoting intestinal peristalsis and alleviating constipation to reduce the secretion of mucinogen granules.

Substance P (SP), somatostatin (SS) and vasoactive intestinal peptides (VIP) are known to lead to regular GI motility [28,29]. The results provided may explain the mechanisms that promote intestinal peristalsis and alleviate constipation as outcomes of UKP feeding targeted at the level of SP. SP is a member of the tachykinin family of neuropeptides, and acts as a neurotransmitter. Investigation over the recent decades has indicated that an unbalanced tachykininergic system may play a role in the pathophysiology of inflammatory bowel disease, and contribute to motor, secretory and immunological disturbances [30]. These findings also call for the need to further study the tachykininergic system, in particular, of the order of SP neurohormonal signaling, for their involvement in GI motility processes such as promoting intestinal peristalsis.

## 4. Conclusions

The results of the present study showed that the microencapsulation and milling process effectively prevented the loss of nutrients and that the ultra-micro grinding of IDF leads to increases in WHC, OHC, SWC and AD. These fundamental changes resulted in a significant increase in GI motility processes and relieved constipation. These results indicated that ultra-micro kiwifruit powder (UKP) has the potential to be developed as a functional ingredient for intestinal health.

## 5. Materials and Methods

### 5.1. Materials

Hayward kiwifruits were purchased from a local market in Yangling. γ-Cyclodextrin (GCD) and monoglyceride (Tongyuan, Qingdao, China) were used as the carrier agents and emulgator. Four phenolic standards (chlorogenic acid, caffeic acid, (−)-epicatechin and gallic acid) were purchased from Sigma-Aldrich (St. Louis, MO, USA). Mouse Substance (SP) Elisa Assay Kit, Mouse Vasoactive Intestinal Peptide (VIP) Elisa Assay Kit and Mouse Somatostatin (SS) Elisa Assay Kit were from Shanghai Rongsheng Biotech (Shanghai, China). All other chemicals were of analytical or HPLC grade. Ultrapure water (18 mΩ) was supplied by a Milli-Q apparatus (Millipore Corp.).

### 5.2. Preparation of Ultra-Micro Kiwifruit Powder (UKP) 

(1) Pretreatment: the fresh and pre-cooled kiwifruit were washed, then peeled and cut into pieces. By adding 0.5 times the weight of kiwifruit and 0.2% sodium tripolyphosphate as a color protector, the crude kiwifruit juice was obtained by filtering with 40 mesh screen after rapid juicing; (2) High-pressure homogenization: the crude kiwifruit juice was homogenized at 20–40 MP under high pressure; (3) Microencapsulation: γ-cyclodextrin, 17% weight of the homogenized juice, was added to the juice, then colloidally grinded several times to obtain grinding fluid; (4) Emulsifying homogenization: the amount of emulsifier glycerol octanoate added to the grinding fluid was 0.5–4% weight of the grinding fluid, and the homogeneous liquid was obtained by homogenizing 1–3 times; (5) Pre-freezing: the homogeneous liquid was pre-frozen at −80 °C for 12 h; (6) Vacuum freeze-drying: the pre-frozen homogeneous liquid was placed in freeze-drying equipment, the temperature of the cold trap was controlled at −70 ~ −72 °C for 36 hours, then microencapsulated dry powder was obtained; (7) Ultrafine comminution: the volume ratio of microencapsulated dry powder and agate ball (diameter 3 mm) was 1:1. The microencapsulated fruit ultrafine powder was obtained by grinding at 300 r/min, 30 min and 1–3 times in the agate grinding bowl of a planetary ball mill [31]. 

Kiwifruit pomace was extracted and mixed with 0.40% alpha-amylase. The ratio of material to liquid was 1:20 (m:v) at 65 °C and pH 6.0 for 80 min; then 0.20% protease was added at 55 °C, pH 3.0 for 60 min [32]; after inactivation at 100 °C, centrifugation was carried out. The filtrate residue was pre-frozen at −80 °C for 12 hours and then placed in freeze-drying equipment. The temperature of the cold trap was controlled at −70 ~72 °C and vacuum freeze-drying was carried out for 36 h. After fully drying, the dietary fiber dry powder of pomace was obtained. The volume ratio of kiwifruit pomace dietary fiber powder to agate ball (diameter 3 mm) was 1:1, which was placed in the agate grinding bowl of the planetary ball mill. The rotational speed was 300 r/min, the grinding time was 45 min and it was ground 3 times. The modified dietary fiber ultra-micro powder of kiwifruit pomace was obtained. UKP was obtained by mixing microencapsulated juice powder with modified dietary fiber of pomace. The full process flow is shown in Figure 5.

### 5.3. The Characterization of UKP Morphology

For scanning electron microscopy (SEM), UKP samples were sputter-coated with gold, and examined with a Hitachi S-3400N scanning electron microscope (Hitachi, Tokyo, Japan) at 5.0 kV. Microscopic images were taken using an inverted optical microscope (Olympus IX71, Olympus, Tokyo, Japan). The native fluorescence was detected using three lasers as excitation sources and appropriate long pass filters (UV argon ion laser). Images were serially obtained by scanning the section with each laser beam in combination with an appropriate emission filter. A series of confocal images were recorded separately for each detection channel. Representative images were chosen from at least three similar images. 

### 5.4. Physicochemical Characteristics of UKP

Water-holding capacity (WHC): 10 mL distilled water was added to 1 g of the dry kiwifruit DF sample. The suspension was homogenized in a vortex for 1 min and left at room temperature for 24 h. After centrifugation at 3000× *g* for 10 min, the supernatant was removed and the residue was weighed. The WHC was expressed as grams of water held by 1 g of dry sample.

Oil-holding capacity (OHC): the OHC was determined in a similar manner as the WHC, but using vegetable oil instead of distilled water. The density of vegetable oil is 0.917 g mL^−1^ at 25 °C. The OHC was expressed as grams of oil held by 1 g of dry sample. 

Swelling capacity (SWC): a weighed level of the dry kiwifruit DF sample (1.0 ± 0.001 g) was added into distilled water (25 mL) in a 50 mL graduated cylinder. The sample was stirred gently with a spatula to eliminate trapped air bubbles. The cylinder was then covered with para film and left undisturbed at room temperature overnight for complete hydration. The volume (mL) occupied by the settled sample was recorded. The SWC was expressed as the volume of swollen sample (mL) per gram of dry sample.

Apparent density (AD): AD refers to the mass per unit volume of material including voids inherent in the material. One gram of the dry Kiwifruit DF was in a calibrated cylinder (15 mL) at room temperature. The apparent density was expressed as g /mL.

### 5.5. HPLC Study of the Polyphenols

About 2 g pretreated UKP was prepared by using microwave-assisted extraction (NJC 03–2; Nanjing Jiequan Microwave Equipment Co., Ltd, Nanjing, China) using aqueous ethanol according to previously reported protocol. The ethanolic extract was evaporated under vacuum, and then re-dissolved in 1 mL methanol. Then, the resulting extract was centrifuged for 15 min at 60,000× *g* (CR21GII; Hitachi Koki Co., Ltd, Toyko, Japan). The extract was then filtered through a membrane filter (0.22 μm) before injection.

Mobile phases used were acidified ultrapure water (0.1% acetic acid, pH 3.2, mobile phase A) and methanol (mobile phase B). The gradient method: 80% A (0–8 min), 65% A (9–12 min), 45% A (13–16 min), 30% A (17–20 min), 20% A (21–30 min), 10% of A (31–34 min) and then washing of the column with 65% A (35–39 min) and lastly, 80% A (40–45 min) was followed. A sample volume of 20 μL was used. The flow rate was maintained at 0.8 mL/min and the wavelengths used for UV-vis detector were 254 nm and 325 nm. The standards used for comparison and identification were chlorogenic acid, caffeic acid, (−)-epicatechin and gallic acid.

### 5.6. Estimation of Ascorbic Acid Content

The 2,6-dichloroindophenol titrimetric method is commonly used in the determination of ascorbic acid. Ascorbic acid content in UKP was estimated by the volumetric method. Five mL of standard ascorbic acid (100 µg/mL) was taken in a conical flask containing 10 mL 4% oxalic acid and was titrated against the 2,6-dichlorophenol indophenols dye. The appearance and persistence of a pink color was taken as the end point. The amount of dye consumed (V1 mL) is equivalent to the amount of ascorbic acid. Five mL of sample (prepared by taking 5 g of juice in 100 mL 4% oxalic acid) was taken in a conical flask with 10 mL of 4% oxalic acid and titrated against the dye (V2 mL). The amount of ascorbic acid was calculated using the formula, ascorbic acid (mg/100 g) = (0.5 mg/V1mL) × (V2/15 mL) × (100 mL/Wt. of sample) × 100.

### 5.7. Estimation of Dietary Fibre Content

Total dietary fiber (TDF), soluble dietary fiber (SDF), and insoluble dietary fiber (IDF) were determined by using an enzymatic–gravimetric method with the fiber assay kit (AOAC 991.43) method [33,34]. In brief, one gram of sample suspended in Mes-Tris buffer was sequentially digested by heat stable α-amylase for 15 min in a boiling water bath, after which protease and amyloglucosidase were added and the mixture was held for 30 min at 60 °C. After filtration, the IDF was recovered from enzyme digestate, dried at 105 °C, and then weighed. SDF in the filtrate was precipitated with ethanol and filtered. The precipitate, referred to as SDF, was dried at 105 °C and weighed. IDF and SDF contents were corrected for residual protein and ash content. The TDF content was the sum of IDF and SDF.

### 5.8. Animals and Treatment

Male C57BL/6J mice (8 months old) were obtained from the Beijing Vital River Laboratory Animal Technology (Beijing, China). Animals were housed in rectangular cages in a controlled atmosphere with a 12-h light/dark cycle and distilled water was provided ad libitum. After 2 weeks of adaptive feeding, the mice were divided into 4 groups (*n* = 10 mice/group): (a) mice fed with AIN-93M commercial basal chow (control group); (b) mice fed with AIN-93M commercial basal chow and with diphenoxylate treatment (5 mg·kg^−1^, i.g.) for 1 week (model group); (c) mice treated with AIN-93M commercial basal chow mixed with freeze dried kiwifruit without peel (2.4g·kg^−1^, B.W.) for 4 weeks and with diphenoxylate treatment (5 mg·kg^−1^, i.g.) for the last 1 week (FKD group); (d) mice treated with AIN-93M commercial basal chow mixed with ultra-micro kiwifruit powder (2.4g·kg^−1^, B.W.) for 4 weeks and with diphenoxylate treatment (5 mg·kg^−1^, i.g.) for the last 1 week (UKP group). The Animal Ethics Committee of Northwest A & F University (Yangling, Shannxi, China) approved the animal protocols. All the animal surgeries were performed under anesthesia.

### 5.9. Fecal Output Studies 

Fecal pellets were collected every 15 min and weighed immediately for a total duration of 1 h for each mouse. Total stool weight was calculated and the result was presented as a percentage of stool weight in vehicle-treated background mice. Dejecta moisture content (DWC) was determined for each group of mice.

### 5.10. GI Transit. Studies In Vivo

Twenty minutes after the administration of diphenoxylate, 200 μL of an Evans blue (5% Evans blue; Sigma-Aldrich, St. Louis, MO, USA; 5% arabic gum; Sigma-Aldrich, St. Louis, MO, USA) marker was gavaged into the stomach. The time taken to detect Evans blue in the feces (in min) was recorded and presented as the whole gut transit time (WGT).

### 5.11. Intestinal Transit. Rates

Briefly, male C57BL/6J mice were fasted overnight with free access to water prior to the experiment. Thirty minutes after the last administration of diphenoxylate or distilled water, 200 μL of an Evans blue (5% Evans blue; Sigma-Aldrich, St. Louis, MO, USA; 5% gum arabic; Sigma-Aldrich, USA) marker was gavaged into the stomach. The mice were killed by exarticulation at 25 min after dye ingestion. The intestine from the pylorus to the caecum was quickly removed and the distance traveled by the Evans blue and the total length of the intestine were measured. The intestinal transit rate was evaluated by the Evans blue propelling ratio, which was calculated as the percentage of the distance traveled by the Evans blue meal relative to the total length of the intestine.

### 5.12. Histology and Immunohistochemistry

The intestine tissue was fixed in 4% (*w*/*v*) paraformaldehyde/PBS and embedded in paraffin. For H&E staining, a part of jejunum was cut off far from the Treitz ligament, about 10 cm, and stained with hematoxylin and eosin. For immunohistochemistry, thick sections were prepared, deparaf-fined, and rinsed three times with PBS (pH 7.4). The Tris-EDTA buffer epitope retrieval method was used in order to retrieve antigen. Then, the slice was treated with 3 % H_2_O_2_ for 10 min to eliminate endogenous peroxidase. The intestine slice was incubated for 20 min in normal goat serum blocking solution to block non-specific binding of immunoglobulin, and the sections were then incubated with primary antibody. The secondary antibody coupled to horseradish peroxidase was incubated at room temperature for 30 min. Intestine sections were washed and visualized by chromogen DAB (DAB kit, Zhongshan Golden Bridge Biotechnology Co. Ltd, Beijing, China) reaction for up to 10 min. Finally, these sections were counterstained with hematoxylin. Intestine sections were dehydrated in ethanol, cleared in xylene, and evaluated on a light microscopy (Olympus, Tokyo, Japan).

### 5.13. Intestinal Neurotransmitter Quantification in Portal Vein Plasma

Mouse substance (SP), mouse omatostatin (SS) and mouse vasoactive intestinal peptide (VIP) levels in portal vein plasma were analyzed by Elisa Assay Kit (Rongsheng Biotech, Shanghai, China).

### 5.14. Statistics

Data are presented as means ± SD of at least three independent experiments. Significant differences between means were determined by one-way ANOVA using GraphPad Prism software. Means were considered to be statistically significant if *p* < 0.05.

## Figures and Tables

**Figure 1 molecules-24-03813-f001:**
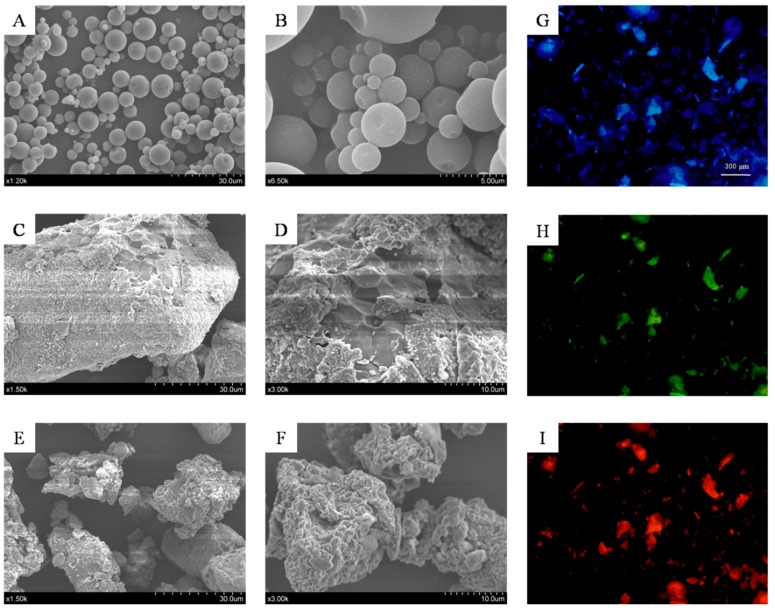
Scanning electron microscopy (SEM) images and fluorescence micrographs of the ultra-micro kiwifruit powder (UKP). SEM: **A**–**B**, SEM images of microencapsulation at 1.2 k and 6.5 k × magnification, respectively; **C**–**D**, SEM images of insoluble dietary fiber (IDF) before the grinding process at 1.5 k and 3.0 k × magnification, respectively; **E**–**F**, SEM images of IDF after the grinding process at 1.5 k and 3.0 k × magnification, respectively. Fluorescence micrographs: images of IDF after the grinding process were obtained by UV argon ion laser; the blue fluorescence (**G**), the green fluorescence (**H**) and the red fluorescence (**I**).

**Figure 2 molecules-24-03813-f002:**
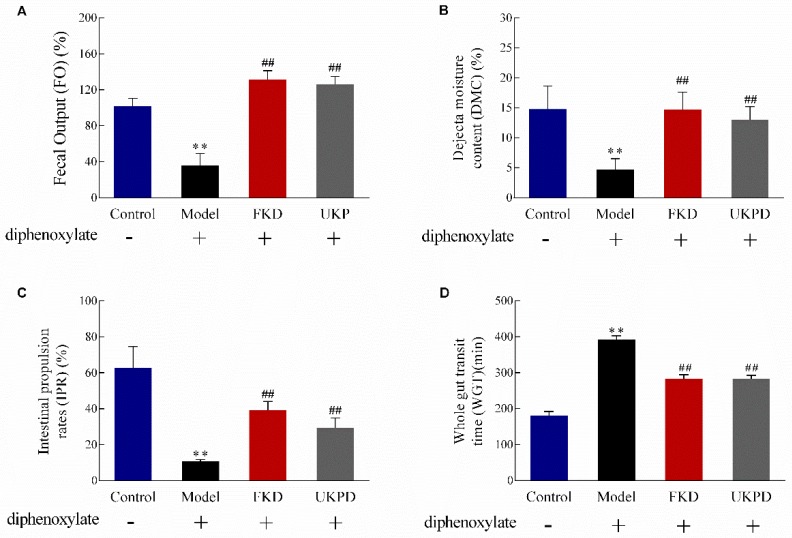
Effect of ultra-micro kiwifruit powder (UKP) and freeze-dried kiwifruit (FKD) on (**A**) fecal output (FO), (**B**) the dejecta moisture content (DWC), (**C**) the intestinal propulsion rates (IPR), and (**D**) the whole gut transit time (WGT) in mice treated with diphenoxylate (5 mg·kg^−1^, i.g.). Data are presented as the mean ± SD, *n* = 8–9 mice per group; two-way ANOVA; Bonferroni post hoc test. ** *p* < 0.001, compared with control group; ^##^
*p* < 0.001, compared with model group.

**Figure 3 molecules-24-03813-f003:**
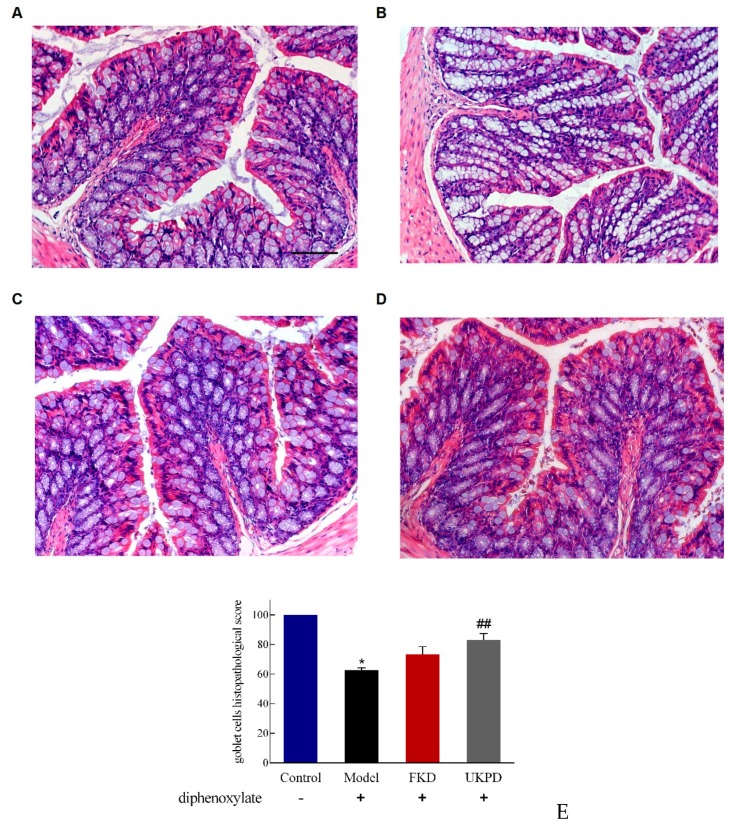
Effects of UKP and FKD on preventing intestinal damage. Representative photomicrographs of hematoxylin and eosin (H&E) staining of jejunum in (**A**) control group, (**B**) model group, (**C**) FKD group, and (**D**) UKPD group. Scale bars: 200 μM. (**E**) Goblet cells histopathological score.

**Figure 4 molecules-24-03813-f004:**
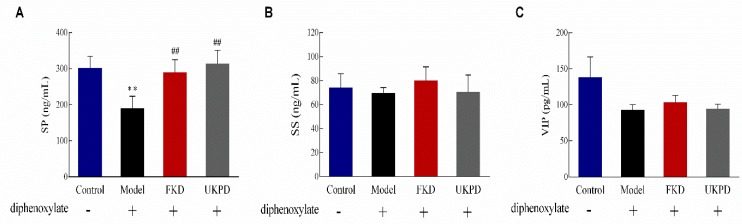
Effects of UKP and FK on (**A**) substance P (SP), (**B**) somatostatin (SS) and (**C**) vasoactive intestinal peptide (VIP) levels in portal vein plasma. Data are presented as the mean ± SD, *n* = 8–9 mice per group; two-way ANOVA; Bonferroni post hoc test. ** *p* < 0.001, compared with control group; ^##^
*p* < 0.001, compared with model group.

**Figure 5 molecules-24-03813-f005:**
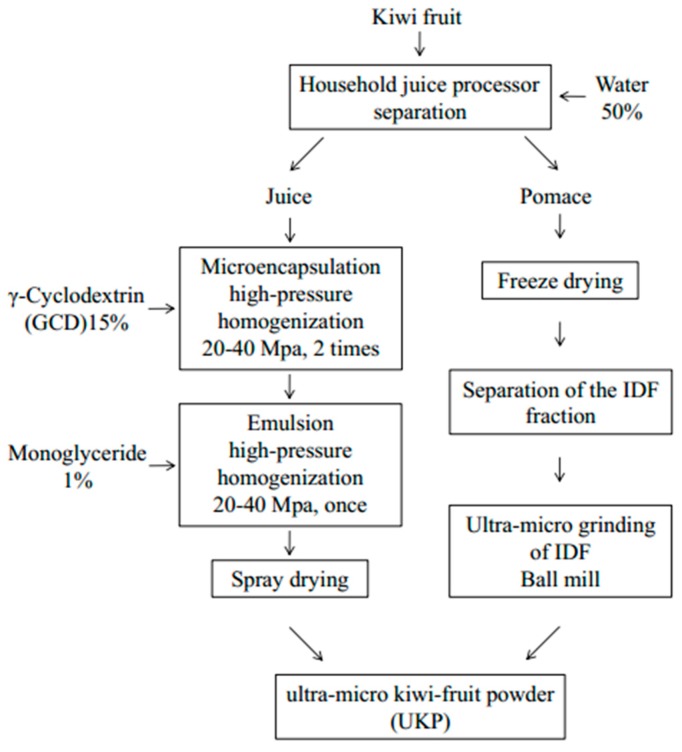
Flow diagram of microencapsulation and grinding process of the ultra-micro kiwifruit powder (UKP) products.

**Table 1 molecules-24-03813-t001:** Water-holding capacity (WHC), oil-holding capacity (OHC), swelling capacity (SWC) and apparent density (AD) of kiwifruit fiber. *^a.^*

	Before Grinding	After Grinding
WHC (g/g)	3.62 ± 0.04a	4.56 ± 0.05b
OHC (g/g)	0.66 ± 0.03a	1.09 ± 0.01b
SWC (mL /g)	6.98 ± 0.11a	7.69 ± 0.19b
AD (g/mL)	0.41 ± 0.006a	0.23 ± 0.009b

*^a^* Values are given as the means ± SD for *n* = 3. Means within the same row followed by different letters are significantly different (*p* < 0.05).

**Table 2 molecules-24-03813-t002:** Effects of ultra-micro grinding on the content of dietary fiber.*^a.^*

	Before Grinding	After Grinding
TDF (%)	81.27 ± 0.73a	84.48 ± 0.21b
SDF (%)	5.31 ± 0.14a	11.87 ± 0.02b
IDF (%)	75.96 ± 11.08a	72.61 ± 3.179b
IDF/SDF	14.3:1	6.1:1

*^a^* Values are given as the means ± SD for *n* = 3. Means within the same row followed by different letters are significantly different (*p* < 0.05).

**Table 3 molecules-24-03813-t003:** Effects of ultra-micro grinding on the content of phenolic and vitamin C.*^a.^*

	Raw Kiwifruit	UKP	Retention (%)
chlorogenic acid	3.227 ± 0.003a	2.833 ± 0.001b	87.8
caffeic acid	0.546 ± 0.009a	0.497 ± 0.002b	91.0
(−)-epicatechin	76.019 ± 0.008a	71.610 ± 0.007b	94.2
Gallic acid	0.506 ± 0.003a	0.443 ± 0.003 b	87.5
Vitamin C	76.84 ± 0. 54a	69.29 ± 0.85 b	90.2

*^a^* Values are given as the means ± SD for *n* = 3. Means within the same row followed by different letters are significantly different (*p* < 0.05).

**Table 4 molecules-24-03813-t004:** Effects of UKP on the body weight and food intake of mice.*^a.^*

Group	Animal no.	Original Body Weight (g)	Final Body Weight (g)	Daily Food Intake (g)
Control	8	16.4 ± 1.3	27.7 ± 1.9	4.9 ± 1.3
Model	8	16.6 ± 0.9	25.1 ± 2.5	4.7 ± 0.7
KFD	9	16.0 ± 0.9	26.6 ± 2.2	5.1 ± 0.8
UKPD	9	16.8 ± 1.1	27.4 ± 1.8	5.0 ± 0.9

*^a^* Values are given as the means ± SD. Compared with control group *p* > 0.05.

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
