# Peer review of "The Manufacturing Process of Kiwifruit Fruit Powder with High Dietary Fiber and Its Laxative Effect"

_molecules, 2019, doi:10.3390/molecules24213813_

Round 1
Reviewer 1 Report
This manuscript aimed to study how kiwifruit powder affects intestinal health under constipation. To answer this question, the authors analyzed the ultrastructure of UKP under SEM and fluorescence microscope and tested the physiochemical properties of UKP. To investigate how UKP affects colonic health in vivo, the authors administrated the mice with UKP supplementation for four weeks and demonstrated that UKP ameliorated constipation that is related to the goblet cell increment. Although This study is an interesting research topic and organized in a good shape, the manuscript does need some work before it can be considered for publication.
The whole manuscript need editing. For example, what is SP stands for in the Abstract. What is the dosage relevant to human consumption? Any quantification data for figure 3? The authors could considering either using WB or staining to show goblet cells at protein level.Author Response
Review 1’s Comments The whole manuscript need editing. For example, what is SP stands for in the Abstract. What is the dosage relevant to human consumption? Any quantification data for figure 3? The authors could considering either using WB or staining to show goblet cells at protein level. Point 1: The whole manuscript need editing. For example, what is SP stands for in the Abstract. Response: Thank you for the suggestion. The term has been modified in the revised manuscript and it is highlighted in red color. Point 2: What is the dosage relevant to human consumption? Response: According to FDA (2005) Guidance for Industry: Estimating the Maximum Safe Starting Dose in Initial Clinical Trials for Therapeutics in Adult Healthy Volunteers, there is a corresponding conversion relationship between the dose intake of mice and human. In the current research, the ultra-micro kiwi-fruit powder (UKP) administered to mice was 2.4 g•kg−1 B.W., and if an adult weight was 70 kg, it should be taken in approximately 20 g UKP. Point 3: Any quantification data for figure 3? The authors could considering either using WB or staining to show goblet cells at protein level. Response: Thank you for your suggestions. According to recent research [1] and our current study, we have provided a goblet cells histopathological score in Figure 3 in the revised manuscript. Reference: [1] Gong Z, Zhao S, Zhou J, et al. Curcumin alleviates DSS-induced colitis via inhibiting NLRP3 inflammasome activation and IL-1β production [J]. Molecular Immunology, 2018, 104:11-19.
Reviewer 2 Report
This manuscript describes the manufacturing process of the whole kiwifruit powder with high dietary fibre and its laxative effects. This work is good but requires much more detailed data.
Major problems:
No information on other components such as sugars, proteins, ions, actinidin content and other polyphenols in the final product. Needs more details such as yield of the preparation, and full compositional analysis of the major components
Most of the laxative studies done with Kiwifruits using whole kiwifruits. they should compare how this compares with whole fruits
Mechanism of the laxative effect of Kiwi is still not known; the author should comment on this issue and the rationale of their method.
The abstract should mention " mice model."
